# Role of 2-[^18^F]FDG as a Radiopharmaceutical for PET/CT in Patients with COVID-19: A Systematic Review

**DOI:** 10.3390/ph13110377

**Published:** 2020-11-10

**Authors:** Salvatore Annunziata, Roberto C. Delgado Bolton, Christel-Hermann Kamani, John O. Prior, Domenico Albano, Francesco Bertagna, Giorgio Treglia

**Affiliations:** 1Nuclear Medicine Unit, IRCSS Regina Elena National Cancer Institute, 00144 Rome, Italy; salvatoreannunziata@live.it; 2Department of Diagnostic Imaging (Radiology) and Nuclear Medicine, University Hospital San Pedro and Centre for Biomedical Research of La Rioja (CIBIR), 26006 Logroño, La Rioja, Spain; rbiolton@gmail.com; 3Department of Nuclear Medicine and Molecular Imaging, Lausanne University Hospital and University of Lausanne, CH-1011 Lausanne, Switzerland; Christel-Hermann.Kamani@chuv.ch (C.-H.K.); John.Prior@chuv.ch (J.O.P.); 4Nuclear Medicine, University of Brescia and Spedali Civili Brescia, 25123 Brescia, Italy; domenico.albano@unibs.it (D.A.); francesco.bertagna@unibs.it (F.B.); 5Clinic of Nuclear Medicine and PET/CT Center, Imaging Institute of Southern Switzerland, Ente Ospedaliero Cantonale, 6500 Bellinzona, Switzerland; 6Health Technology Assessment Unit, Academic Education, Research and Innovation Area, Ente Ospedaliero Cantonale, 6500 Bellinzona, Switzerland

**Keywords:** fluorodeoxyglucose, FDG, PET, COVID-19, SARS-CoV-2, infection, systematic review

## Abstract

Some recent studies evaluated the role of fluorine-18 fluorodeoxyglucose (2-[^18^F]FDG) as a radiopharmaceutical for positron emission tomography/computed tomography (PET/CT) imaging in patients with Coronavirus Disease (COVID-19). This article aims to perform a systematic review in this setting. A comprehensive computer literature search in PubMed/MEDLINE and Cochrane library databases regarding the role of 2-[^18^F]FDG PET/CT in patients with COVID-19 was carried out. This combination of key words was used: (A) “PET” OR “positron emission tomography” AND (B) “COVID” OR “SARS”. Only pertinent original articles were selected; case reports and very small case series were excluded. We have selected 11 original studies of 2-[^18^F]FDG PET/CT in patients with COVID-19. Evidence-based data showed first preliminary applications of this diagnostic tool in this clinical setting, with particular regard to the incidental detection of interstitial pneumonia suspected for COVID-19. To date, according to evidence-based data, 2-[^18^F]FDG PET/CT cannot substitute or integrate high-resolution CT to diagnose suspicious COVID-19 or for disease monitoring, but it can only be useful to incidentally detect suspicious COVID-19 lesions in patients performing this imaging method for standard oncological and non-oncological indications. Published data about the possible role of 2-[^18^F]FDG PET/CT in patients with COVID-19 are increasing, but larger studies are warranted.

## 1. Introduction

Coronavirus Disease (COVID-19) is an infectious acute respiratory syndrome caused by the virus SARS-CoV-2 (Severe Acute Respiratory Syndrome Coronavirus 2) able to enter the human cells through the angiotensin-converting enzyme 2 (ACE2) receptor, like the SARS virus [1]. COVID-19 mainly affects the lower respiratory tract and causes several flu-like symptoms. In severe cases, COVID-19 may cause interstitial pneumonia, which can evolve in acute respiratory distress syndrome and subsequently death. Real-time reverse polymerase chain reaction (RT-PCR) is a useful test for the diagnosis of SARS-CoV-2 infection, with several limitations due to possible false-negative results [1].

COVID-19 suspicious patients are usually assessed with radiological investigations, and high-resolution chest computer tomography (CT) is the preferred tool [2]. Usually, in the early phases, single or multiple ground glass opacities, nodules, or plaques may appear. With the disease progression, the lesions may increase, occupying most of the lungs [2]. Furthermore, several authors showed an ever-growing incidental pulmonary findings suggestive of COVID-19 highlighted in the CT component of the predominantly oncological positron emission tomography (PET)/CT studies of patients without a known SARS-CoV-2 infection, COVID-19 diagnosis, or symptoms [3].

Fluorine-18 fluorodeoxyglucose (2-[^18^F]FDG) is a radiopharmaceutical representing a radiolabelled glucose analogue that can be used to evaluate the glucose metabolism (Figure 1). Its cellular uptake by cells is proportional to their glucose consumption. After the uptake of this radiopharmacetical through the glucose transporters located in the cell membrane, it is phosphorylated by hexokinase [4]. As the cells involved in inflammatory response have high glycolytic activity (with high expression of glucose transporters and increased hexokinase activity) then they are able to take up 2-[^18^F]FDG and therefore 2-[^18^F]FDG PET/CT may be able to identify sites of inflammation and infection [4].

Several recently published studies have evaluated the role of 2-[^18^F]FDG PET/CT in patients with COVID-19. This article aims to perform a systematic review according to related guidelines [5,6] to investigate the possible role of 2-[^18^F]FDG PET/CT in this setting.

## 2. Results

From the comprehensive computer literature search of PubMed/MEDLINE and Cochrane databases, 169 records were screened. In total, 11 studies were selected and retrieved in full-text version according to the predefined inclusion criteria [7,8,9,10,11,12,13,14,15,16,17]. Conversely, 158 records were excluded because case reports or small case series (*n* = 51), reviews, editorials, letters (*n* = 39) or because they were outside the field of interest of this review (*n* = 68). No additional articles were found checking the references of the selected articles. Figure 2 shows the flowchart of study selection and search results. The characteristics of the selected studies are presented in Table 1 and Figure 3 and summarized here below.

All the studies were published in 2020 during the COVID-19 outbreak from European authors (from Italy, Spain, United Kingdom, France, and Monaco) and all except one of them are retrospective observational studies.

In most of the studies, 2-[^18^F]FDG PET/CT was performed in patients evaluated for standard indications (mostly oncological indications) and incidental findings suspicious of SARS-CoV-2 infection or COVID-19 at whole-body 2-[^18^F]FDG PET/CT were described [7,8,9,10,12,13,14,15,16,17]. Only in one study 2-[^18^F]FDG PET/CT was performed in COVID-19 patients to assess the inflammatory status at the presumed peak of the inflammatory phase [11].

Most of the included studies reported the rate of incidental findings suspicious of SARS-CoV-2 infection or COVID-19 at 2-[^18^F]FDG PET/CT performed for standard indications. This rate largely varied among the included studies from 2.1% to 16.2% of all whole body 2-[^18^F]FDG PET/CT scans [7,8,9,12,13,14,15,16,17], based on the prevalence of SARS-CoV-2 infection or COVID-19 during the examination period in a specific country or region [11,12,16]. A significant association of the appearance of imaging findings suspicious for COVID-19 on 2-[^18^F]FDG PET/CT with gender, sex, age, or ethnicity was not demonstrated [12]. Only a percentage of suspicious COVID-19 at 2-[^18^F]FDG PET/CT was finally confirmed by further examinations (e.g., RT-PCR or serological testing) [7,8,9,10,12,13,14,15,16,17].

As reported by all the included studies, the 2-[^18^F]FDG PET/CT findings suspicious for SARS-CoV-2 infection or COVID-19 are mainly CT signs of interstitial pneumonia (e.g., ground-glass opacities, crazy paving patterns, and/or consolidations) with involvement of both lungs in most of the cases [7,8,9,10,11,12,13,14,15,16,17].

In cases of pulmonary CT findings suspicious for SARS-CoV-2 infection or COVID-19 at 2-[^18^F]FDG PET/CT, areas of pneumonia were 2-[^18^F]FDG avid in most of the cases but with heterogeneous 2-[^18^F]FDG uptake [7,8,9,10,11,12,13,14,15,16,17].

In patients with pulmonary CT findings suspicious for SARS-CoV-2 infection or COVID-19 at 2-[^18^F]FDG PET/CT, the radiopharmaceutical uptake (measured as maximal standardized uptake value—SUVmax) from pulmonary consolidations was usually higher than radiopharmaceutical uptake from ground-glass opacities and crazy paving patterns [8]. One study found that 2-[^18^F]FDG uptake of pulmonary CT abnormalities, in particular in cases of consolidation areas, were significantly lower in patients with a final diagnosis of COVID-19 compared to those who were negative for COVID-19 [8]. Conversely, another study did not find a significant difference of 2-[^18^F]FDG uptake in ground glass pulmonary infiltrates between COVID-19 patients and controls [17].

Some studies reported that the prevalence of incidental pulmonary findings suspicious for interstitial pneumonia by COVID-19 at 2-[^18^F]FDG PET/CT was higher when compared to 2-[^18^F]FDG PET/CT scans of the same period one year earlier, without significant differences in 2-[^18^F]FDG uptake [7,12,13,16].

Beyond the lungs, increased 2-[^18^F]FDG uptake in patients with suspicious COVID-19 may be frequently detected in lymph nodes (in particular mediastinal lymph nodes) and less frequently in extra-thoracic sites [7,10,11,12,13,16,17].

Interestingly, cases with known COVID-19 and negative 2-[^18^F]FDG PET/CT findings are described, even if less frequently [10].

The only prospective study aimed to assess the inflammatory status by 2-[^18^F]FDG PET/CT in patients with known COVID-19 demonstrated that there was no correlation between pulmonary inflammatory status at 2-[^18^F]FDG PET and chest CT evolution or short-term clinical outcome [11].

## 3. Discussion

2-[^18^F]FDG is a radiolabelled glucose analogue and the most common PET radiopharmaceutical used in oncology or for detecting inflammatory diseases. An increased uptake of this radiopharmaceutical is expected in inflammatory/infectious sites and tumor lesions due to the usually high glycolytic metabolism of activated inflammatory cells as well as tumor cells [4]. An advantage of 2-[^18^F]FDG PET as a functional imaging is the possible identification of metabolically active diseases in the early stages, which usually precedes morphological changes appreciated in conventional imaging techniques such as CT. Therefore, by detecting the active phase of an infectious or inflammatory condition, it may be theoretically possible to diagnose and monitor the inflammatory/infectious disease progression [4].

We have performed a systematic review of the literature to evaluate whether a possible role of 2-[^18^F]FDG as a radiopharmaceutical for PET/CT in patients with suspicious, known, or unknown COVID-19 may be hypothesized. Searching the literature, we found a plethora of case reports and very small case series describing incidental findings of COVID-19 in patients performing 2-[^18^F]FDG PET/CT for common indications (mainly for oncological purposes). We have excluded these case reports and small case series from our analysis due to their low value on the quality of evidence point of view. Conversely, we have restricted the inclusion criteria only to original articles including more than five patients with suspicious or known COVID-19 evaluated by 2-[^18^F]FDG PET/CT because these articles provide more reliable information compared to case reports.

However, most of the studies selected about our topic of interest were observational and retrospective studies mainly focused on incidental detection of COVID-19 by 2-[^18^F]FDG PET/CT (*n* = 10) [7,8,9,10,12,13,14,15,16,17] and only one study was a prospective analysis of patients with known COVID-19 aimed to assess the inflammatory status by 2-[^18^F]FDG PET/CT and the possible correlation with CT evolution and short-term clinical outcome [11].

First of all, our review clearly demonstrates that the rate of incidental findings suspicious for COVID-19 in asymptomatic patients undergoing 2-[^18^F]FDG PET/CT is significant, ranging from 2 to 16% of 2-[^18^F]FDG PET/CT examinations performed during the COVID-19 outbreak as reported in the included studies. This rate of incidental findings is also related to the prevalence of COVID-19 in different countries and regions [7,8,9,12,13,14,15,16,17]. Due to the relevant heterogeneity among the included studies and some biases in particular about the reference standard (Figure 3), we have not performed a meta-analysis to calculate the pooled prevalence of incidental findings suspicious for COVID-19 at 2-[^18^F]FDG PET/CT, because this pooled analysis would be not appropriate [5,6,18].

The lungs are the most frequent sites of incidental findings suspicious for COVID-19 at 2-[^18^F]FDG PET/CT [7,8,9,10,11,12,13,14,15,16,17] and this observation is quite obvious considering the pulmonary tropism of SARS-CoV-2 [1,2]. Most of the pulmonary incidental findings suspicious for COVID-19 at 2-[^18^F]FDG PET/CT are detected by using the CT component, which may allow to recognize ground-glass opacities, crazy paving patterns, and/or consolidations [7,8,9,10,11,12,13,14,15,16,17]. Nevertheless, it should be underlined that the co-registered CT component of PET/CT is often a low-dose CT (useful for attenuation correction and for anatomical localization of PET findings) not performed on a breath-hold. The diagnostic quality of this low-dose CT is not entirely equivalent to dedicated chest CT having a lower detection rate for smaller pulmonary lesions; furthermore, motion artefacts generated on a free-breathing CT thorax can artificially create ground glass opacities particularly in the lower pulmonary lobes [7,8,9,10,11,12,13,14,15,16,17]. Improving the diagnostic quality of the co-registered CT component of the PET/CT is possible following EANM guidelines [19], by trimming certain aspects of the PET/CT procedure. Including in the protocol a deep-inspiration thoracic CT scan, which is not used for attenuation correction or for PET/CT fusion, but instead to assess the lung parenchyma with thinner slices (2.5 mm), is very useful for studying the lung parenchyma, especially for comparing with previous and/or future studies [19]. In summary, the need for incorporating this CT protocol to the PET/CT should be considered given the relevance of precisely evaluating lung parenchyma, especially in patients who have not had a thoracic CT scan recently.

A recent published meta-analysis focused on the diagnostic performance of CT in the diagnosis of COVID-19 demonstrated that observation of ground glass opacities and other radiological features of viral pneumonia at CT scan had optimum performance in detection of COVID-19 [20]. However, it is suggested to make the final diagnosis based on both CT scan and RT-PCR, as none of these diagnostic tests are reliable alone [20].

In this scenario, the increased 2-[^18^F]FDG uptake related to the increased metabolic activity of active inflammatory lesions may help to identify COVID-19 lesions at 2-[^18^F]FDG PET/CT [7,8,9,10,11,12,13,14,15,16,17]. The relatively lower 2-[^18^F]FDG uptake observed in pulmonary lesions of COVID-19 patients compared to those of COVID-19 negative patients could be explained a trend toward lower rates of CT consolidation patterns in COVID-19 positive than in COVID-19 negative patients. Indeed, consolidation patterns were associated with much higher SUVmax values when compared with ground-glass opacities or crazy paving patterns, which include a non-cellular component and therefore are less likely to take up 2-[^18^F]FDG [8]. However, as a significant overlap of SUVmax values was demonstrated in pulmonary lesions from COVID-19 positive and COVID-19 negative patients, 2-[^18^F]FDG PET/CT alone cannot be used to discriminate between COVID-19 and other pulmonary infections [7,8,9,10,11,12,13,14,15,16,17]. Moreover, 2-[^18^F]FDG PET/CT cannot be used to exclude COVID-19 due to the possibility of a negative 2-[^18^F]FDG PET/CT in COVID-19 patients. Furthermore, the sensitivity of 2-[^18^F]FDG PET/CT in detecting post-infective pulmonary lesions that are no longer hypermetabolic is potentially limited [10].

Beyond the lungs, increased 2-[^18^F]FDG uptake in patients with suspicious COVID-19 may be frequently detected in lymph nodes (in particular mediastinal lymph nodes) and less frequently in extra-thoracic sites [7,10,11,12,13,16,17]. Notably, lymph nodal 2-[^18^F]FDG uptake is probably linked to cytokine-mediated inflammation rather than to direct viral infection, considering the absence of ACE2 (a functional receptor for SARS-CoV-2) on lymphoid cells [11,17].

Overall, literature data have demonstrated an increased prevalence of 2-[^18^F]FDG PET/CT abnormalities evocative of a pulmonary infection in asymptomatic patients during the COVID-19 outbreak. Therefore, as for other organs [21], PET/CT reading physicians should consider and recognize pulmonary incidental findings, and in particular potential COVID-19 related features, at 2-[^18^F]FDG PET/CT. These findings should be reported to the referring physicians immediately for appropriate action, as the incidental detection of COVID-19 pneumonia in early stages may help to prevent further spread of the virus [7,8,9,10,12,13,14,15,16,17,22,23,24,25].

Even if one article included in our systematic review has reported the possible utility of 2-[^18^F]FDG PET/CT to monitor the inflammatory status in patients with known COVID-19, and even if 2-[^18^F]FDG PET/CT may be feasible in patients with confirmed SARS-CoV-2 infection when taking adequate protective measures [10,25], the current available evidence are scarce and not sufficient to justify the use of 2-[^18^F]FDG PET/CT in patients with suspicious or known COVID-19 in the clinical practice and further research is needed before this can be eventually considered [3]. It should be noted that, in comparison to CT, 2-[^18^F]FDG PET/CT is a complex procedure requiring more human resources and a long stay in the investigation unit, bearing a certain risk of infectious disease spreading for the health personnel. Moreover, we should be aware about the radiation issue since CT and 2-[^18^F]FDG PET/CT are both radiating procedure [3].

Regarding future studies, other radiopharmaceuticals beyond 2-[^18^F]FDG exploring different metabolic pathways or receptor status have been evaluated in patients with COVID-19 but, to date, only preclinical studies or case reports (with low quality of evidence) are currently available [26,27,28,29,30,31,32] and more studies are needed in this regard.

The main limitation of our systematic review is that a limited number of articles were included; therefore, more studies with an adequate number of patients are needed to better understand the possible role of 2-[^18^F]FDG PET/CT in COVID-19 patients or with SARS-CoV-2 infection.

## 4. Materials and Methods

PubMed/MEDLINE and Cochrane library databases were screened on 30th September 2020 to perform a comprehensive computer literature search according to international guidelines (including PRISMA) [5,6] to find published original studies on the role of 2-[^18^F]FDG PET/CT in patients with COVID-19. A search algorithm based on the combination of these terms was used: (A) “PET” OR “positron emission tomography” AND (B) “COVID” OR “SARS”. The literature search was performed by two authors independently (S.A. and G.T.).

Original studies including more than five patients with suspicious or known COVID-19 evaluated by 2-[^18^F]FDG PET/CT during the SARS-CoV-2 outbreak were eligible for inclusion. Case reports and small case series (*n* ≤ 5 cases with suspicious or known COVID-19), reviews, editorials, letters, and articles outside the field of interest of this review (including preclinical studies) were excluded. No language or time restrictions were used.

Titles and abstracts of the retrieved articles were reviewed by two authors (S.A. and G.T.) applying the inclusion criteria mentioned above. References of the eligible articles were also checked to find potential additional articles. For each selected article information was collected about authors, country, type of study, period of study, type of patients evaluated by 2-[^18^F]FDG PET/CT, number of patients with known or suspicious SARS-CoV-2 infection or COVID-19 at 2-[^18^F]FDG PET/CT. The main findings of the selected studies were briefly described and risk of bias assessment of the studies was performed through the QUADAS-2 tool [5,6].

## 5. Conclusions

To date, according to evidence-based data, 2-[^18^F]FDG PET/CT cannot substitute or integrate high-resolution CT to diagnose suspicious SARS-CoV-2 infection or COVID-19 or for disease monitoring, but it can only be useful to incidentally detect suspicious COVID-19 lesions in patients performing this imaging method for standard oncological and non-oncological indications. Published data about the possible role of 2-[^18^F]FDG PET/CT in patients with COVID-19 are increasing, but larger studies are warranted.

## Figures and Tables

**Figure 1 pharmaceuticals-13-00377-f001:**
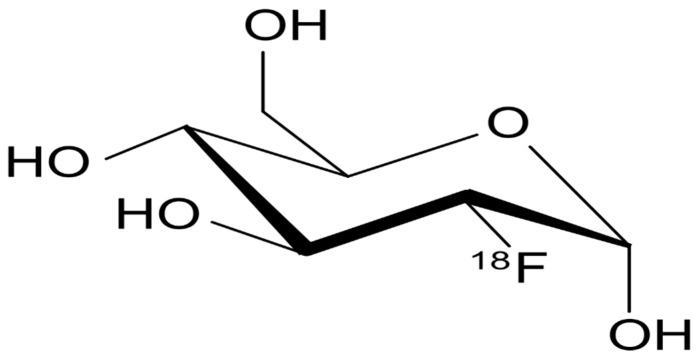
Chemical structure of 2-[^18^F]FDG.

**Figure 2 pharmaceuticals-13-00377-f002:**
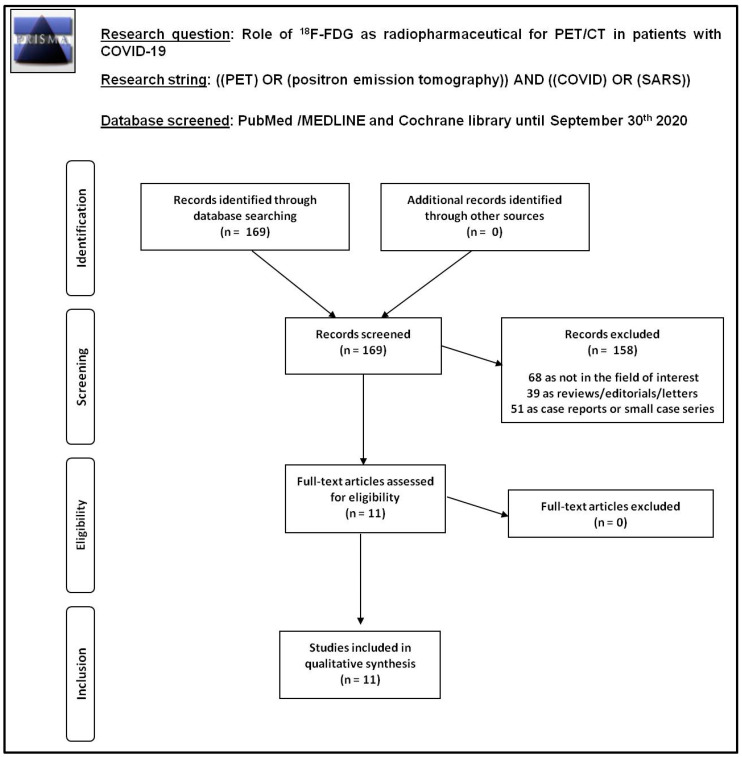
Flowchart of study selection and search results.

**Figure 3 pharmaceuticals-13-00377-f003:**
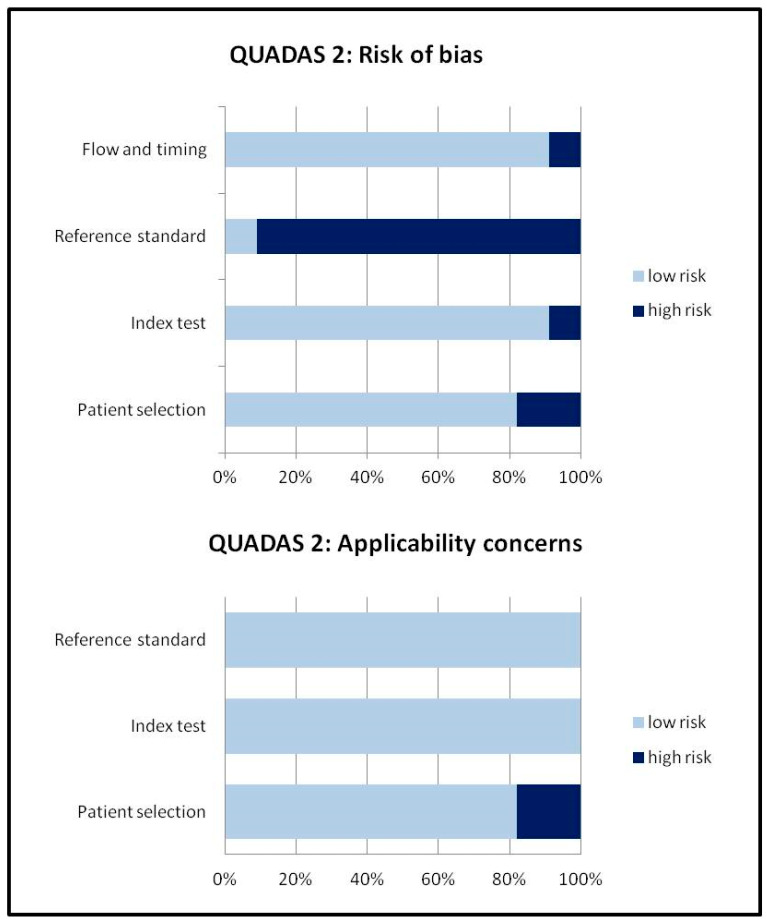
Quality assessment of the included studies according to QUADAS-2 tool.

**Table 1 pharmaceuticals-13-00377-t001:** Characteristics of included studies on the role of 2-[^18^F]FDG PET/CT in patients with COVID-19.

First Author	Country	Type of Study	Period of the Study	Type of Patients Evaluated by 2-[^18^F]FDG PET/CT	Patients with Suspicious (Confirmed) SARS-CoV-2 Infection or COVID-19 Evaluated by 2-[^18^F]FDG PET/CT	Rate of Incidental Findings Suspicious of COVID-19 at 2-[^18^F]FDG PET/CT	Other Endpoints
Albano [7]	Italy	R	March 2020	Asymptomatic patients evaluated for standard indications	6 (4)	9.2% (6/65)	Radiological and metabolic findings in patients with suspicious COVID-19
Bahloul [8]	France	R	March–May 2020	Asymptomatic patients evaluated for standard indications	22 (11)	2.5% (22/884)	Radiological and metabolic findings in patients with suspicious COVID-19
Cabrera Villegas [9]	Spain	R	March–April 2020	Asymptomatic patients evaluated for standard indications	7 (5)	5.3% (7/132)	Radiological and metabolic findings in patients with suspicious COVID-19
Charters [10]	United Kingdom	R	2020 (months N.A.)	Asymptomatic or known COVID-19 patients evaluated for standard indications	6 (4)	N.A.	Radiological and metabolic findings in patients with suspicious or known COVID-19
Dietz [11]	Monaco	P	March–May 2020	Known COVID-19 patients to evaluate the inflammatory status	(13)	N.A.	Radiological and metabolic findings in patients with known COVID-19 and correlation between inflammatory status and CT evolution or short-term outcome
Halsey [12]	United Kingdom	R	March–April 2020	Asymptomatic or symptomatic patients evaluated for standard indications	26 (1)	16.2% (26/160)	Radiological and metabolic findings in patients with suspicious COVID-19
Maurea [13]	Italy	R	February–April 2020	Asymptomatic patients evaluated for standard indications	26 (0)	8.7% (26/299)	Radiological and metabolic findings in patients with suspicious COVID-19
Mucientes Rasilla [14]	Spain	R	March–April 2020	Asymptomatic patients evaluated for standard indications	11 (5)	8.5% (11/129)	Radiological and metabolic findings in patients with suspicious COVID-19
Olivari [15]	Italy	R	April 2020	Asymptomatic patients evaluated for standard indications	7 (6)	4.1% (7/172)	Radiological and metabolic findings in patients with suspicious COVID-19
Setti [16]	Italy	R	December 2019–May 2020	Asymptomatic patients evaluated for standard indications	24 (4)	4.2% (24/575)	Radiological and metabolic findings in patients with suspicious COVID-19
Wakfie-Corieh [17]	Spain	R	February–May 2020	Asymptomatic patients evaluated for standard indications	23 (14)	2.1% (23/1079)	Radiological and metabolic findings in patients with suspicious COVID-19

Legend: N.A. = not available; P = prospective; R = retrospective.

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
