# Peer review of "Role of 2-[18F]FDG as a Radiopharmaceutical for PET/CT in Patients with COVID-19: A Systematic Review"

_pharmaceuticals, 2020, doi:10.3390/ph13110377_

Round 1

Reviewer 1 Report

The authors have done a search on “PET” OR “positron emission tomography” AND “COVID” or “SARS”.

The description on how they did the search which papers they left out and why and that they have look at the reference in the papers the included to fine other articles which there search could have overlooked is nicely described and easy to follow. However, they don’t tell us which date they did the search.

Their search only gave 169 articles, 158 records were exclude 68 because they were outside field of interest (PS: there is a mistake in figure 1, where you write that there were 254 is outside the field of interest), then 39 is exclude because they are reviews, editorials, letters and finally 51 studies are exclude because they are too small.

This leave only 11 studies to be analyzed. 10 of the 11 studies operates with what they call “suspicious cases” as well as confirmed COVID cases, only one  studies  only include confirmed COVID cases.

By including the suspicious case in the review I think you introduce a big uncertainty/ unknown into the study. On the other hand, I’m not sure why you don’t look at the 51 studies just because they are small?

I believe this review would be significantly better at that by wait to more papers have been submitted (which I suspect happens all the time). I also think you should divide you analysis up in confirmed cases and “suspicious cases” (if you want to include them at all). It would be of interest to know for how long the patients have had COVID, so if you could include those data to it would be appreciated.

Author Response

Comment: The description on how they did the search which papers they left out and why and that they have look at the reference in the papers the included to fine other articles which there search could have overlooked is nicely described and easy to follow. However, they don’t tell us which date they did the search.

Reply: We have reported in the revised manuscript that the literature search was performed on September 30th 2020.

Comment: Their search only gave 169 articles, 158 records were excluded 68 because they were outside field of interest (PS: there is a mistake in figure 1, where you write that there were 254 is outside the field of interest), then 39 is exclude because they are reviews, editorials, letters and finally 51 studies are exclude because they are too small.

Reply: We have modified the Figure 1 due to the mistake indicated by the reviewer. We have provided a revised Figure 1. 

Comment: This leave only 11 studies to be analyzed. 10 of the 11 studies operates with what they call “suspicious cases” as well as confirmed COVID cases, only one  studies  only include confirmed COVID cases. By including the suspicious case in the review I think you introduce a big uncertainty/ unknown into the study. On the other hand, I’m not sure why you don’t look at the 51 studies just because they are small?

Reply: we agree with the reviewer that having a combination of suspicious and confirmed cases may be a bias. We have already reported this bias in the quality assessment of the included articles (category: reference standard). Furthermore, we have also reported in the discussion that "Due to the relevant heterogeneity among the included studies and some biases in particular about the reference standard (Figure 2), we have not performed a meta-analysis to calculate the pooled prevalence of incidental findings suspicious for COVID-19 at 2-[18F]FDG PET/CT, because this pooled analysis would be not appropriate".
About the case reports we have excluded them from the analysis because these articles are low quality rticles according to the evidence pyramid. We have focused our review on original articles with an adequate number of patients to have more reliable information compared to case reports as reported in the discussion of the revised manuscript.

Comment: I believe this review would be significantly better at that by wait to more papers have been submitted (which I suspect happens all the time). I also think you should divide you analysis up in confirmed cases and “suspicious cases” (if you want to include them at all). It would be of interest to know for how long the patients have had COVID, so if you could include those data to it would be appreciated.

Reply: We have performed this review for providing timely information to the readers on a "hot topic" as several articles on this topic were published in the last months. We surely will perfom an update of this analysis in the next years when more articles will be available, but we believe that a systematic review at this point on this hot topic could be much appreciated by the readers.

About suspicious and confirmed cases, it is difficult to extract the information related to these two groups by the included articles. Nevertheless, we have already divided one article focused on known cases to the rest of articles that include mainly suspected cases. About the information on how long the patients have had COVID this is not reported in most of the articles therefore we have not reported this information in our review which is mainly focused on the role of FDG-PET/CT in COVID-19 patients.

Reviewer 2 Report

The  manuscript "Role of 2-[18F]FDG as a radiopharmaceutical for PET/CT in patients with COVID-19: a systematic review", by Annunziata et al present a review of selected publications.

As the authors point out it could be of importance, considering the corona virus situation at the moment, to be aware of incidentally detected or suspicious lesions with [18F]FDG related to a covid-19 infection.

The methodology used is adequate and the conclusion drawn in line with the results. 

Author Response

We thank the Reviewer for having appreciated our manuscript.

Reviewer 3 Report

This is a systematic review based on published recent results in only 11 studies, therefore the statistic is poor; I would recommend authors to acknowledge this limitation in the discussion. In the same line, when the selected studies are with a low number of subjects (>5) and include other pathological condition (mainly oncological),  if may be useful to mention, also in the discussion chapter, some of the results of case reports - to confirm or, on the contrary to question the main findings of the review and the general conclusion.

Author Response

Comment: This is a systematic review based on published recent results in only 11 studies, therefore the statistic is poor; I would recommend authors to acknowledge this limitation in the discussion. In the same line, when the selected studies are with a low number of subjects (>5) and include other pathological condition (mainly oncological),  if may be useful to mention, also in the discussion chapter, some of the results of case reports - to confirm or, on the contrary to question the main findings of the review and the general conclusion.

Reply: According to the reviewer's comment, we have added in the discussion of the revised manuscript that "The main limitation of our systematic review is that a limited number of articles were included; therefore, more studies with an adequate number of patients are needed to better understand the possible role of 2-[18F]FDG PET/CT in COVID-19 patients or with SARS-CoV-2 infection".

Case reports are excluded from the analysis due to their low quality.

Reviewer 4 Report

The manuscript by Annunziata et al. entitled “Role of 2-[18F]FDG as a radiopharmaceutical for PET/CT in patients with COVID-19: a systematic review.” presents a systematic analysis of publications on the applicability of [18F]FDG/PET-CT for the diagnosis and monitoring of COVID-19.

High sensitivity and functional nature of PET together with worldwide accessibility of [18F]FDG make  [18F]FDG/PET-CT an attractive diagnostic tool. It is rather certain that numerous attempts are made around the world to implement it into clinical routine, and in the context of Covid-19 pandemic it is especially urgent. Thus, it is of utmost importance for both academic and clinical communities to receive the overview of the initial experience.

The analysis of the literature and publication inclusion criteria are very well structured and clearly presented. The manuscript is concise and well-organized.   

Author Response

We thank the Reviewer for having appreciated our systematic review.

Round 2

Reviewer 1 Report

My objections have not been met